# Associations between big five personality traits, facets, and sexual fantasies

Emily Cannoot[1], Amy C. Moors[2,3], William J. Chopik[1*]

1 Department of Psychology, Michigan State University, East Lansing, Michigan, United States of America, 2 Department of Psychology, Chapman University, Orange, California, United States of America, 3 The Kinsey Institute, Indiana University, Bloomington, Indiana, United States of America

* chopikwi@msu.edu

## Abstract

The present study investigated associations between Big Five personality traits, their facets, and the frequency and variety of sexual fantasies in a sample of 5,225 adults ($M = 58.30$ years old; 56.5% men). People high in conscientiousness and agreeableness report fewer sexual fantasies across exploratory, intimate, impersonal, and sadomasochistic domains; these effects were primarily driven by respectfulness and responsibility facets. Negative emotionality, particularly the depression facet, was associated with more frequent sexual fantasizing. Extraversion and open-mindedness showed minimal relationships with sexual fantasy frequency. These results underscore the importance of personality dimensions in understanding sexual thoughts, emphasizing the heterogeneity of sexual fantasies across individuals. Having a broader appreciation of the heterogeneity of sexual experiences can foster more inclusive approaches to sexual well-being in psychological research and clinical practice.

## Introduction

Sexual fantasies are common yet rarely discussed phenomena, often because of stigma and embarrassment around experiencing them [1,2]. Because they are less often discussed, there is some ignorance about how common sexual fantasies are, who is most likely to have them, and if types of fantasies vary according to people's psychological traits. Among the little work done examining these questions, most has focused on broad indicators of personality while overlooking facet-level information on personality traits. Examining specific traits might provide a more complete description of "who" has sexual fantasies. The current study examined associations between Big Five personality traits and facets and the frequency and variety of sexual fantasies in a sample of over 5,000 adults.

**Data availability statement:** All data, syntax, and materials are available from OSF at https://osf.io/c9pzj/.

**Funding:** The author(s) received no specific funding for this work.

**Competing interests:** The authors have declared that no competing interests exist.

## Sexual fantasies and their role in people's lives

In the broadest sense, sexual fantasies are defined as "any mental imagery that is sexually arousing or erotic to the individual" [1]. Common taxonomies characterize sexual fantasies as multidimensional and generally varying several domains including romantic, experimental, personal/impersonal, masochistic, and force dominance-related fantasies [3]. Although there is some uncertainty about the exact origin of sexual fantasies, there is at least some acknowledgement that they are at least partially shaped by social and societal processes [1,4]. Most people admit to fantasizing about sex and enjoy doing so, regardless of their age, gender identity, or sexual orientation [1,5]. Nevertheless, there are a few demographic differences. Specifically, men tend to fantasize more than women, but there is some evidence that women may be more likely to engage in fulfilling these fantasies, particularly more romantic/intimate fantasies, although actual engagement is less commonly studied [3,6]. Sexual fantasies are also more common among younger adults [7].

When trying to assess how sexual fantasies are associated with psychological characteristics, part of the difficulty in doing so is a historical framing of sexual fantasies as problematic or as reflective of psychopathology. In fact, many studies have focused on associations between sexual fantasies and exclusively antisocial behavior like aggressive tendencies or sex offending [8, 9]. However, the experience of sexual fantasies is so heterogeneous that merely knowing a person's sexual fantasies does not necessarily map on to their mental health characteristics as seen in correlational studies and reviews of sexual fantasies [9,10]. In other words, non-aggressive people could have aggressive sexual fantasies; interpersonally dominant people might have submissive sexual fantasies; introverted people might have voyeuristic fantasies.

Given that sex is one of the more underappreciated sources of well-being in people's lives [11], there is evidence to suggest that more frequent sexual fantasizing is associated with relationship promoting behaviors (e.g., verbalizing affection) and positive relationship outcomes [12,13]. Health and well-being benefits accompany positive and fulfilling relationships, so researchers have turned toward more deliberately included considerations of sex and sexuality in the study of health and well-being across the lifespan [7,14]. Given that sexual fantasies are both common and might have some positive effects on people's happiness and relationships, what personality characteristics predict endorsement of various sexual fantasies?

## Personality traits and sexual fantasy

Personality traits reflect people's characteristic patterns of thoughts, feelings, and behaviors. Currently, there is a general consensus regarding the Big Five as a framework for understanding variation in people's personalities [15]. The Big Five personality traits are comprised of extraversion, agreeableness, conscientiousness, negative emotionality (i.e., neuroticism), and open-mindedness (i.e., openness to experience) [16]. Among the many life outcomes linked with personality, neuroticism is associated with more sexual dissatisfaction, negative emotions about sex, and symptoms of sexual dysfunction; openness is associated with more liberal attitudes toward sex

[17]. Although the Big Five framework is the most commonly studied set of higher-order traits, there have been relatively few studies examining their associations with sexual fantasies (and no studies examining personality *facets* and sexual fantasies, from what we could find). In fact, there is little information *at all* about how sexual fantasizing is associated with any psychological traits (especially non-psychopathological traits).

This lack of research is apparent in one of the largest meta-analyses on personality and sexual behavior to date [17]. Among the 137 studies included in that paper, the authors could only identify between two to five effect sizes that implicated sexual fantasies, depending on the trait being studied. They found that people higher in openness ($r = 24$), extraversion ($r = .10$), and neuroticism ($r = .13$) tended to fantasize more often, although each of these studies collapsed across different types of sexual fantasies. Studies looking at more isolated traits (e.g., sensation seeking) that could reasonably be subsumed under the Big Five are also somewhat common [18,19]. In the work conducted since that 2018 meta-analysis, a more complicated picture of the association between personality and sexual fantasies has emerged, sometimes contradicting the meta-analysis. For example, extraverts and open-minded individuals are more likely to fantasize about extra-dyadic sex and consensual non-monogamy; people high in neuroticism were less likely to report these types of fantasies [10,20]. Yet, other research suggests that introverts might have more sexual fantasies than extraverts [21]. People high in conscientiousness or agreeableness were less likely to have aggressive sexual fantasies than those low in these traits [8,22]. Highly neurotic people tend to have both more positive *and* negative sexual thoughts, including violent fantasies [4,21,23]. People open to experience (i.e., open mindedness) tend to sexually daydream more often than those less open to experience [24]. Yet, other studies find few or no associations between personality and sexual fantasies of various types [25]. We revisited this question in a large sample of adults who reported on their personalities and how much they engaged in a variety of sexual fantasies.

One additional contribution our study makes to the literature is examining associations between sexual fantasies and personality *facets*. Personality facets are lower order traits, subsumed by the Big Five, that provide more specific information about people's personalities [26,27]. Integrating facets into an analysis can reveal more complete insights into who is most likely to have certain sexual fantasies. For example, if extraverts have more sexual fantasies [20], is it primarily driven by their sociability, their assertiveness, or the energy level (all facets of extraversion)? Likewise, if conscientious people are less likely to have aggressive fantasies [8], it is likely because of the respectfulness and compassionate facets of the trait rather than the trust facet per se. In addition to focusing on links between people's personality facets and sexual fantasies, we ran supplementary analyses in which we controlled for the covariation among the Big Five traits in their associations with sexual fantasies. Given the mixed results from previous research, and the lack of research about facets, we treated all analyses as exploratory.

## Method

The present exploratory study was not preregistered. All data, syntax, and materials can be found at https://osf.io/c9pzj/?view_only=b9672f35521f483c90a84701dfb105fc This study was carried out in accordance with the recommendations of Michigan State University's Institutional Review Board (IRB# x16-1291e) and run online with informed consent being secured from all participants (by clicking a next arrow; documentation requirement waived). Data were analyzed anonymously.

## Participants and procedure

Participants comprised of 5,225 internet respondents recruited from Qualtrics Panels who took part in an online study about close relationships in exchange for $10. Data were collected from March 15, 2021 through October 21, 2021. They ranged in age from 18 to 94 ($M_{age} = 58.30$, $SD = 15.93$). The sample was 56.5% men, 43.2% women, and .3% who described themselves as another gender. The sample was 87.7% White, 3.5% Asian, 3.2% Hispanic/Latinx, 3.1% Black/African American, and 2.5% other races/ethnicities.

Given the scope of the present study focused on close relationships, all participants were either married (94.5%) or dating (5.5%). Mean relationship length was 29.68 years ($SD = 16.93$). The sample was relatively sexually active, with 68.6% of participants having sex at least once per month. The inclusion criteria were that people were currently in a relationship and at least 18 years old. We collected as many participants as funding allowed (see below for power analysis though). We did not exclude any participants, and missing data were handled via listwise deletion.

## Measures

**Big five personality traits.** The Big Five Personality traits were measured using the short-form version of the Big Five-Inventory-2 [28]. The questionnaire contains 30 items (6 for each trait), and individuals respond to each statement according to how well it describes them on a scale ranging from 1 (*disagree strongly*) to 5 (*agree strongly*). Mean composites were computed for extraversion (e.g., "I am someone who is outgoing, sociable;" $\alpha = .71$, $M = 3.27$, $SD = .72$), agreeableness (e.g., "I am someone who is compassionate, has a soft heart;" $\alpha = .81$, $M = 3.85$, $SD = .71$), conscientiousness (e.g., "I am someone who is persistent, works until the task is finished;" $\alpha = .80$, $M = 3.93$, $SD = .73$), negative emotionality (e.g., "I am someone who worries a lot;" $\alpha = .84$, $M = 2.37$, $SD = .84$), and open-mindedness (e.g., "I am someone who is fascinated by art, music, or literature;" $\alpha = .73$, $M = 3.37$, $SD = .73$).

Within each of the broader Big Five personality traits are three facets—smaller, more specific descriptors of personality. We also calculated the means for three facets (measured with two items each) of extraversion (sociability ($\alpha = .63$), assertiveness ($\alpha = .65$), and energy level ($\alpha = .68$)), agreeableness (compassion ($\alpha = .61$), respectfulness ($\alpha = .57$), and trust ($\alpha = .61$)), conscientiousness (organization ($\alpha = .74$), productiveness ($\alpha = .59$), and responsibility ($\alpha = .46$)), negative emotionality (anxiety ($\alpha = .67$), depression ($\alpha = .67$), and emotional volatility ($\alpha = .63$)), and open-mindedness (aesthetic sensitivity ($\alpha = .53$), intellectual curiosity ($\alpha = .45$), and creative imagination ($\alpha = .56$)).

**Sexual fantasies.** Sexual fantasies were assessed with the Sexual Fantasies Questionnaire [29], which asked participants to indicate how often they fantasized about each of 40 themes on a scale ranging from 1 (*never*) to 6 (*daily*). In addition to an overall composite of sexual fantasizing ($\alpha = .97$, $M = 1.73$, $SD = .90$), the responses can be subsumed into four 10-item composite scales: exploratory sexual fantasies (e.g., "participating in an orgy;" $\alpha = .94$, $M = 1.52$, $SD = .94$), intimate sexual fantasies (e.g., "making love outdoors in a romantic setting;" $\alpha = .93$, $M = 2.45$, $SD = 1.21$), impersonal sexual fantasies (e.g., "watching others have sex;" $\alpha = .91$, $M = 1.58$, $SD = .90$), and sadomasochistic sexual fantasies (e.g., "being forced to do something;" $\alpha = .95$, $M = 1.37$, $SD = .89$).

## Analytical approach

We began by running bivariate associations between personality traits, facets, and the different sexual fantasy outcomes (i.e., a composite total and each of the four composite scales). Then, to account for the covariation, we ran follow-up linear regressions that included all personality traits, age, and gender. From these models, partial correlations were extracted. For these models that featured control variables, we ran two models, one for the Big Five traits and another for the (15) facets. These were done separately because the broader Big Five are made up of the facets and including them all in one model would introduce multicollinearity problems. This information is presented in Table 1 in the form of partial correlations.

## Results

With our sample size of 5,225 people, we could detect effects as small as $r = .039$ with 80% power at $\alpha = .05$. Because of the large sample size, many small correlations (even those slightly smaller than $r = .039$) were statistically significant at $p < .05$ (all $rs > |.03|$ and $rs_p > |.027|$). Thus, we primarily focus on the magnitude of the correlations and highlight the largest and most robust associations.

**Table 1. Bivariate and Partial Correlations between Personality and Sexual Fantasies.**

| Construct | Fantasy Total | | Exploration | | Intimate | | Impersonal | | Sadomasochistic | |
|---|---|---|---|---|---|---|---|---|---|---|
| | Bivariate | Partial | Bivariate | Partial | Bivariate | Partial | Bivariate | Partial | Bivariate | Partial |
| Extraversion | −0.001 | 0.058 | −0.033 | 0.037 | 0.081 | 0.085 | −0.034 | 0.037 | −0.041 | 0.041 |
| Agreeableness | −0.228 | −0.083 | −0.243 | −0.098 | −0.123 | −0.015 | −0.260 | −0.110 | −0.233 | −0.089 |
| Conscientiousness | −0.265 | −0.131 | −0.278 | −0.141 | −0.158 | −0.068 | −0.270 | −0.119 | −0.286 | −0.150 |
| Negative Emotionality | 0.171 | 0.038 | 0.173 | 0.023 | 0.100 | 0.051 | 0.180 | 0.042 | 0.189 | 0.016 |
| Open Mindedness | −0.005 | 0.023 | −0.033 | 0.008 | 0.080 | 0.075 | −0.042 | 0.004 | −0.052 | −0.026 |
| Sociability | 0.004 | 0.061 | −0.007 | 0.057 | 0.040 | 0.051 | −0.016 | 0.053 | −0.008 | 0.058 |
| Assertiveness | 0.004 | −0.031 | −0.020 | −0.035 | 0.073 | 0.001 | −0.011 | −0.033 | −0.045 | −0.047 |
| Energy Level | −0.011 | 0.086 | −0.046 | 0.064 | 0.067 | 0.087 | −0.047 | 0.064 | −0.039 | 0.081 |
| Compassion | −0.201 | −0.021 | −0.212 | −0.024 | −0.111 | −0.002 | −0.226 | −0.029 | −0.207 | −0.026 |
| Respectfulness | −0.258 | −0.085 | −0.271 | −0.093 | −0.149 | −0.034 | −0.276 | −0.085 | −0.272 | −0.103 |
| Trust | −0.115 | 0.028 | −0.126 | 0.024 | −0.052 | 0.026 | −0.148 | 0.006 | −0.108 | 0.045 |
| Organization | −0.187 | 0.005 | −0.194 | 0.005 | −0.126 | −0.008 | −0.183 | 0.016 | −0.193 | 0.010 |
| Productiveness | −0.205 | −0.034 | −0.221 | −0.037 | −0.106 | −0.017 | −0.216 | −0.035 | −0.230 | −0.038 |
| Responsibility | −0.283 | −0.055 | −0.294 | −0.061 | −0.168 | −0.016 | −0.289 | −0.054 | −0.306 | −0.071 |
| Anxiety | 0.070 | −0.081 | 0.075 | −0.087 | 0.032 | −0.025 | 0.078 | −0.085 | 0.081 | −0.101 |
| Depression | 0.384 | 0.276 | 0.380 | 0.270 | 0.263 | 0.171 | 0.382 | 0.274 | 0.402 | 0.290 |
| Emotional Volatility | 0.185 | 0.003 | 0.182 | −0.005 | 0.123 | 0.013 | 0.190 | 0.007 | 0.196 | −0.007 |
| Aesthetic Sensitivity | 0.039 | 0.055 | 0.020 | 0.054 | 0.086 | 0.055 | 0.004 | 0.043 | 0.012 | 0.037 |
| Intellectual Curiosity | −0.005 | −0.020 | −0.025 | −0.021 | 0.064 | 0.005 | −0.030 | −0.025 | −0.050 | −0.041 |
| Creative Imagination | −0.051 | −0.010 | −0.078 | −0.026 | 0.037 | 0.022 | −0.078 | −0.016 | −0.092 | −0.027 |

*Note.* Correlations $r > |.03|$ and $r_p > |.027|$ are significant at $p < .05$. More intensely green cells correspond to more strongly positive correlations. More intensely red cells correspond to more strongly negative correlations. Yellowish cells fall in between these extremes and contain smaller correlations.

Table 1 presents a heat map of the associations between Big Five personality traits and sexual fantasy frequencies (both bivariate and after partialling out the other traits, age, and gender). More intensely green colors correspond to more positive correlations. More intensely red colors correspond to more negative correlations. Varying shades of yellow correspond to values between these intensely positive and negative correlations and are typically smaller and closer to zero. Bivariate and partial correlations greater than |.03| were significant at $p = .05$.

As seen in Table 1, the most consistent predictor of sexual fantasies was conscientiousness. Higher levels of conscientiousness were associated with a lower frequency of all four types of sexual fantasies (exploration, intimate, impersonal, and sadomasochistic). The next most consistent predictor was agreeableness; higher levels of agreeableness associated with a lower frequency of most types of fantasies. Extraverts tended to fantasize about all four types of sexual fantasies more, but only *after* the covariation between traits was controlled for. People with higher levels of negative emotionality fantasized about all four types of sexual fantasies. Open-mindedness was mostly unrelated to fantasies, particularly after controlling for covariation with other traits, age, and gender. One consistent pattern that emerged is that associations between personality traits and sexual fantasy were reduced after controlling for covariation between the traits and demographics (i.e., comparing the bivariate and partial correlation columns in Table 1).

In the bottom portion of Table 1 (run in separate models than the higher-order traits), the facet most largely correlated with higher levels of frequent fantasizing in all categories was depression (a facet of negative emotionality), such that people with more depressive personalities tended to fantasize about all four types of sexual fantasies. Respectfulness (for agreeableness) and responsibility (for conscientiousness) had the strongest negative associations with each type of sexual fantasy, although the magnitude of these associations was reduced dramatically in the covariate analysis. Most of the

remaining facet associations had correlations smaller than $r=|.20|$, and many associations either became non-significant or even switched signs (albeit these associations were near-zero) after controlling for demographic characteristics and the covariation with the other facets. One surprising finding is that the open-mindedness facets, including creative imagination, were largely unrelated to any type of sexual fantasy, which one might expect because open people tend to fantasize about both sexual and non-sexual things more often [24,30].

## Discussion

We examined associations between sexual fantasies and the Big Five personality traits and facets. People high in agreeableness and conscientiousness reported less frequent sexual fantasies, both overall and for specific types of fantasies (e.g., exploration, intimate, impersonal, sadomasochistic). People high in negative emotionality tended to fantasize more. Measuring facets proved worthwhile in that we were able to discern the parts or facets of these traits that are related to sexual fantasy. Specifically, those with depressive personalities reported more frequent sexual fantasies. Those who were high in respectfulness and responsibility reported less frequent sexual fantasies. One implication of the current work is that individual differences in personality might be useful in predicting variation in sexual fantasy frequencies, although they are not wholly redundant with each other (and some associations are relatively small or modest). Knowing these associations further advances the predictive power of personality while showing that variation in sexual fantasies is common.

### Big five personality and sexual fantasies

Despite most people experiencing sexual fantasies, sexual fantasies are relatively understudied or framed as pathological [1,9]. Research shows the utility in approaching the study of sexual fantasies in a value-neutral way has revealed that sexual fantasies are often linked to positive relationship and life outcomes and are also common across the lifespan [7,12–14]. There are relatively few studies examining associations between psychological predictors, like the Big Five personality traits [18]. However, those studies largely showed that people higher in agreeableness and conscientiousness are less likely to report sexual fantasizing, people higher in openness might be more likely to have sexual fantasies in some contexts, and people high in negative emotionality were more likely to have a variety of sexual fantasies [see 10, 18, 20, 23, 24].

We were able to replicate some of these findings but not others. Indeed, people higher in agreeableness or conscientiousness tended to report less frequent sexual fantasies. Although negative emotionality was associated with more frequent sexual fantasies (and sexual fantasies of different types), these associations dropped to near-zero (and in many cases non-significance). Openness to experience was largely unrelated to sexual fantasies (except a positive association with intimate sexual fantasies). Extraversion had a small positive association with sexual fantasies, but this mostly only occurred in the models with covariates (and the signs of the associations changed directions). Thus, the main take-aways for the broader trait level are that agreeableness and conscientiousness are negatively related to sexual fantasy frequencies; extraversion and negative emotionality are positively related to sexual fantasy frequencies—although it mostly depends on whether covariates are controlled for; and openness is mostly unrelated to sexual fantasies. These findings were seen across different types of fantasies, whether they were exploratory, intimate, impersonal, or sadomasochistic in nature. Agreeableness and conscientiousness are associated with norm endorsement, harm prevention, and traditionalism [31,32]. Thus, it seems reasonable to assume that they might be less likely to engage in fantasies that are non-traditional, bridge social norms, or simulate consensual aggression.

The findings with respect to negative emotionality were a bit harder to interpret. People high in negative emotionality are more likely to focus on negative stimuli, perseverate, and have negative thoughts overall [33,34]. They are also more likely to experience various types of disgust, including sexual disgust [23,35]. These observations together might be why they are more likely to think about certain types of sexual fantasies (e.g., sadomasochistic), but this does not explain why they are prone to more ostensibly positive or neutral fantasies as well. However, our study merely examined the frequency

of sexual fantasies and not their utility. In other words, some researchers have suggested that people high in negative emotionality might engage in sexual fantasies as an emotion regulation tool to compensate for negative mood [22]. Indeed, when sexual fantasies are operationalized in terms of valence (e.g., positive or negative fantasies, people high in negative emotionality tend to have both positive *and* negative sexual thoughts and fantasies [22,23]. Thus, people high in negative emotionality tend to fantasize more overall, such that they have both positive and negative sexual fantasies.

## Facets and fantasy

Examining associations between Big Five personality facets (or subcomponents of the broader traits) enabled a more focused examination. Knowing more about these subcomponents has significantly advanced our understanding of personality-outcome associations as has been the case in intra- and interpersonal contexts [26,36,37]. Although some of the Big Five traits were associated with sexual fantasies, it is possible that there are subcomponents of those larger traits that can give us a clue for why people fantasize more or less.

Indeed, respectfulness (for agreeableness) and responsibility (for conscientiousness) were the drivers of the negative associations seen among their higher-order traits. What this reveals is that agreeable people may be less likely to sexually fantasize because of their respect for norms and others (hence the less common sadomasochistic fantasies) and not that tender feelings about others or perceptions of trust are holding them back from doing so (particularly in the covariate analyses). Likewise, people high in responsibility are likely to moralize themselves and others and avoid sexual thinking (and maybe sex) altogether [17]. What the facet analysis also tells us is that people who are more or less organized (or more or less productive) are no more or less likely to sexually fantasize. The large association with depression (a facet of negative emotionality) can also reveal some reasons why negative emotionality was positively associated with sexual fantasies. Specifically, the fact that depression was strongly related to sexual fantasy—and that anxiety and emotional volatility were not—provides support for the possible emotional regulatory function of sexual fantasies [22,23]. In other words, it is not the case that emotionally volatile people or anxious people were more likely to sexually fantasize (if anything, anxious people were slightly less likely to fantasize in some models). Rather, those more prone to negative emotionality were also those most likely to sexually fantasize, possibly as a way to have more positive cognitions [23]. The fact that open-mindedness was largely unrelated to sexual fantasies was a bit surprising, particularly given past research [24,30]. There is a precedent for null results in this literature [25], and future research can examine if these results might be attributable to the way open-mindedness is operationalized [38].

## Limitations and future directions

This study had many strengths, including having a large sample of adults providing information on their personality facets and a variety of sexual fantasies. Nevertheless, some limitations must be discussed.

First, we relied entirely on self-report data, which may be prone to bias [39], particularly when discussing sensitive topics like sexual fantasies, which many people find embarrassing or uncomfortable to discuss [2]. Worth noting, associations between personality and sexual fantasies and thoughts tend to transcend differences in social desirability [22]. Also, the anonymity of our questionnaire likely also reduced these concerns. Nevertheless, future research can investigate ways to circumvent the social desirability effects that accompany self-reports about sexual fantasies. We also hope that this future work recruits a more diverse sample, including those from non-US countries and among people with different relationship statuses (e.g., single people, partnered people, non-monogamous people).

Second, our data were cross-sectional and was merely a snapshot of people's sexual fantasies and how they were related to personality traits. Stability information on sexual fantasies is relatively rare, but they do tend to fluctuate over the course of a few weeks and months [40,41] and tend to differ across the lifespan [7]. Future research can follow participants longitudinally to determine if personality traits prospective development of sexual fantasies. Further, it is possible that personality traits and sexual fantasies might co-develop together over time such that fantasies might become more

or less common (or different types of fantasies might become more salient) as people's personalities or life circumstances change.

## Conclusion

The present study demonstrated that personality meaningfully relates to patterns of sexual fantasy. At the broad trait level, conscientiousness and agreeableness were associated with lower fantasy frequency, whereas neuroticism predicted greater fantasy engagement. Importantly, examining personality facets revealed more specific psychological correlates of fantasy, with higher depressive tendencies and lower responsibility and respectfulness emerging as the most consistent predictors. This facet-level approach offers a more nuanced understanding of who fantasizes and how. This extends past work that has relied primarily on global traits and provides a preliminary assessment of how sexual fantasies and related to novel and more specific characteristics. Because sexual fantasies are common yet highly variable across individuals, identifying personality correlates may help clinicians and educators support more informed, sex-positive conversations that acknowledge differences in sexual thought and expression. Future work should continue refining these associations and examine whether personality dynamics predict changes in fantasy over time or across relational contexts.

## Author contributions

**Conceptualization:** Amy C. Moors, William J. Chopik.

**Data curation:** Amy C. Moors, William J. Chopik.

**Formal analysis:** Emily Cannoot, William J. Chopik.

**Investigation:** Amy C. Moors, William J. Chopik.

**Methodology:** Amy C. Moors, William J. Chopik.

**Project administration:** William J. Chopik.

**Resources:** William J. Chopik.

**Supervision:** William J. Chopik.

**Visualization:** Emily Cannoot, William J. Chopik.

**Writing – original draft:** Emily Cannoot, William J. Chopik.

**Writing – review & editing:** Emily Cannoot, Amy C. Moors, William J. Chopik.

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
