## [Decision Letter · Decision Letter 0]

28 May 2025

Dear Dr. Chopik,

Thank you for submitting your manuscript to PLOS ONE. After careful consideration, we feel that it has merit but does not fully meet PLOS ONE’s publication criteria as it currently stands. Therefore, we invite you to submit a revised version of the manuscript that addresses the points raised during the review process.

**One reviewer supports the manuscript as is, while the other suggests several minor but important revisions that need to be addressed. These include clarifying the theoretical framework, elaborating on demographic variables, and improving the reporting of limitations. Please revise the manuscript accordingly.**

We look forward to receiving your revised manuscript.

Kind regards,

Vittorio Lenzo

Academic Editor

PLOS ONE

**Journal Requirements:**

1. When submitting your revision, we need you to address these additional requirements. Please ensure that your manuscript meets PLOS ONE's style requirements, including those for file naming. The PLOS ONE style templates can be found at https://journals.plos.org/plosone/s/file?id=wjVg/PLOSOne_formatting_sample_main_body.pdf and https://journals.plos.org/plosone/s/file?id=ba62/PLOSOne_formatting_sample_title_authors_affiliations.pdf 2. Please note that your Data Availability Statement is currently missing the repository name. If your manuscript is accepted for publication, you will be asked to provide these details on a very short timeline. We therefore suggest that you provide this information now, though we will not hold up the peer review process if you are unable. 3. Please include your full ethics statement in the ‘Methods’ section of your manuscript file. In your statement, please include the full name of the IRB or ethics committee who approved or waived your study, as well as whether or not you obtained informed written or verbal consent. If consent was waived for your study, please include this information in your statement as well.

Reviewers' comments:

Reviewer's Responses to Questions

**Comments to the Author**

1. Is the manuscript technically sound, and do the data support the conclusions?

Reviewer #1: Yes

Reviewer #2: Yes

2. Has the statistical analysis been performed appropriately and rigorously?

Reviewer #1: Yes

Reviewer #2: Yes

3. Have the authors made all data underlying the findings in their manuscript fully available?

Reviewer #1: Yes

Reviewer #2: Yes

4. Is the manuscript presented in an intelligible fashion and written in standard English?

Reviewer #1: Yes

Reviewer #2: Yes

**Reviewer #1:** The paper addresses an important topic related to stress and resilience. However, some areas need improvement. Below is some feedback for improvement.

- Add literature review for second objective association between nurse’s demographic characteristics and both stress and resilience (eg: gender, age)

- State what is the theory being used in this study.

- Methodology- What is the inclusion and exclusion criteria for the targeted sample.

- Add more demographic characteristics in the table related to the study (eg: age)

- Limitation, hanging sentence. “As such”

- Add implication and conclusion related to the second objective of the study

**Reviewer #2:**  This interesting study of 5,225 adults examined the links between Big Five personality traits and sexual fantasies. High conscientiousness and agreeableness, driven by the facets of respectfulness and responsibility, were associated with fewer fantasies across various domains. Neuroticism, particularly the depression facet, correlated with more frequent fantasizing, while extraversion and open-mindedness showed minimal impact. The well-conducted study’s findings reveal how personality shapes sexual thoughts, aiding clinicians in fostering sex-positive therapy. It highlights the diversity of sexual fantasies and supports inclusive approaches to sexual well-being.

**Do you want your identity to be public for this peer review?** For information about this choice, including consent withdrawal, please see our Privacy Policy

Reviewer #1: No

Reviewer #2: No

---

## [Author Response · Author response to Decision Letter 1]

13 Jun 2025

Response to Reviewers (see appended letter for correct formatting)

We would like to thank the editor and reviewers for their thoughtful comments on the manuscript. We very much appreciate the constructive feedback and believe that the manuscript has improved significantly as a result of their suggestions. Below, we report how each of the reviewer issues was addressed and the corresponding changes to the manuscript. The reviewer comments are non-bolded, and our responses are bolded. We are happy to make any additional changes recommended by the Editor.

Editor

One reviewer supports the manuscript as is, while the other suggests several minor but important revisions that need to be addressed. These include clarifying the theoretical framework, elaborating on demographic variables, and improving the reporting of limitations. Please revise the manuscript accordingly.

Thank you for handling our manuscript. In evaluating the reviewers’ comments, we elected to add details related to the inclusion/exclusion criteria in the Method (p 7) and to add a brief remark on the Implications of the study (see pg 12-13).

Reviewer #1

The paper addresses an important topic related to stress and resilience. However, some areas need improvement. Below is some feedback for improvement.

- Add literature review for second objective association between nurse’s demographic characteristics and both stress and resilience (eg: gender, age)

- State what is the theory being used in this study.

- Methodology- What is the inclusion and exclusion criteria for the targeted sample.

- Add more demographic characteristics in the table related to the study (eg: age)

- Limitation, hanging sentence. “As such”

- Add implication and conclusion related to the second objective of the study

We thank the reviewer for taking the time to provide a review. In discussions with the Editor, they partially left it up to our discretion for how to integrate comments from this review. After reviewing the comments, we ultimately decided to add information related to our inclusion/exclusion criteria (see p 7) and about the implications of our study (see pages 12-13). The other remarks fell a bit beyond the purview of our study or did not correspond to our manuscript, so we elected not to make those particular changes.

Reviewer #2

This interesting study of 5,225 adults examined the links between Big Five personality traits and sexual fantasies. High conscientiousness and agreeableness, driven by the facets of respectfulness and responsibility, were associated with fewer fantasies across various domains. Neuroticism, particularly the depression facet, correlated with more frequent fantasizing, while extraversion and open-mindedness showed minimal impact. The well-conducted study’s findings reveal how personality shapes sexual thoughts, aiding clinicians in fostering sex-positive therapy. It highlights the diversity of sexual fantasies and supports inclusive approaches to sexual well-being.

We thank the reviewer for their positive evaluation of our manuscript!

---

## [Decision Letter · Decision Letter 1]

6 Nov 2025

Associations between Big Five Personality Traits, Facets, and Sexual Fantasies

PLOS ONE

Dear Dr. Chopik,

Thank you for submitting your manuscript to PLOS ONE. After careful consideration, we feel that it has merit but does not fully meet PLOS ONE’s publication criteria as it currently stands. Therefore, we invite you to submit a revised version of the manuscript that addresses the points raised during the review process.

https://journals.plos.org/plosone/s/submission-guidelines#loc-laboratory-protocols . Additionally, PLOS ONE offers an option for publishing peer-reviewed Lab Protocol articles, which describe protocols hosted on protocols.io. Read more information on sharing protocols at https://plos.org/protocols?utm_medium=editorial-email&utm_source=authorletters&utm_campaign=protocols .

We look forward to receiving your revised manuscript.

Kind regards,

Vilfredo De Pascalis

Academic Editor

PLOS ONE

Journal Requirements:

Additional Editor Comments:

Both Reviewers and I think that the manuscript can be accepted for publication. However, as suggested by Reviewer #3, they are invited to (1) check whether all the references used are appropriately related to the topic of this article; (2) provide a more detailed explanation of the figure to enhance clarity and interpretability; (3) to strengthen the concluding statements to summarize the study’s findings succinctly and clearly articulate the novelty of the work.

Thus, I invite the authors to resubmit their revised manuscript as soon as possible, in line with the points outlined above, for acceptance.

Reviewer's Responses to Questions

**Comments to the Author**

Reviewer #3: All comments have been addressed

Reviewer #4: All comments have been addressed

2. Is the manuscript technically sound, and do the data support the conclusions?

Reviewer #3: Yes

Reviewer #4: Yes

3. Has the statistical analysis been performed appropriately and rigorously?

Reviewer #3: Yes

Reviewer #4: Yes

4. Have the authors made all data underlying the findings in their manuscript fully available?

Reviewer #3: Yes

Reviewer #4: Yes

5. Is the manuscript presented in an intelligible fashion and written in standard English?

Reviewer #3: Yes

Reviewer #4: Yes

Reviewer #3: This article may be corrected as follows :

I have carefully examined the improvements made by the researchers to conclude that this article can be accepted and published by the PLOS ONE Journal. However, researchers must make improvements to relate the references used better to the topic of this article.

The figure should be explained in more detail to make it easy to understand. The conclusions compiled in paragraphs may need to be improved to explain the results of this study briefly and affirm the existence of novelty in this article.

*** I found the strengths of this article, but there are still weaknesses that need to be fixed.

This article is Acceptable in the PLOS ONE Journal.

Reviewer #4: The study presents the results of original research.

Results reported have not been published elsewhere.

Experiments, statistics, and other analyses are performed to a high technical standard and are described in sufficient detail.

Conclusions are presented in an appropriate fashion and are supported by the data.

The article is presented in an intelligible fashion and is written in standard English.

The research meets all applicable standards for the ethics of experimentation and research integrity.

The article adheres to appropriate reporting guidelines and community standards for data availability.

**Do you want your identity to be public for this peer review?** For information about this choice, including consent withdrawal, please see our Privacy Policy

Reviewer #3: **Yes:** Muhammad Ali Equatora

Reviewer #4: **Yes:** Ana Carina Henriques Teodósio Moisão

---

## [Author Response · Author response to Decision Letter 2]

16 Dec 2025

Response to Reviewers

We would like to thank the editor and reviewers for their thoughtful comments on the manuscript. We very much appreciate the constructive feedback and believe that the manuscript has improved significantly as a result of their suggestions. Below, we report how each of the reviewer issues was addressed and the corresponding changes to the manuscript. The reviewer comments are non-bolded, and our responses are bolded. We are happy to make any additional changes recommended by the Editor.

Editor

Both Reviewers and I think that the manuscript can be accepted for publication. However, as suggested by Reviewer #3, they are invited to (1) check whether all the references used are appropriately related to the topic of this article; (2) provide a more detailed explanation of the figure to enhance clarity and interpretability; (3) to strengthen the concluding statements to summarize the study’s findings succinctly and clearly articulate the novelty of the work.

Thus, I invite the authors to resubmit their revised manuscript as soon as possible, in line with the points outlined above, for acceptance.

Thank you for serving as editor for this paper, expediting its review, and for the positive evaluation! We have now removed references that were tangentially related to the point being made near it in the text, inserted a stand-alone paragraph describing the table/figure, and revised the conclusion to more succinctly state the study findings while expanding our discussion on the novelty of this work and its importance for motivating more constructive conversations across domains. These changes are all detailed below in response to each reviewer comment.

Reviewer #3

1.) I have carefully examined the improvements made by the researchers to conclude that this article can be accepted and published by the PLOS ONE Journal.

However, researchers must make improvements to relate the references used better to the topic of this article.

Thank you for your positive evaluation of our manuscript. We now see that it might some of the citations might have too-distant a connection to what they are supporting in the text. We have now gone through and either added additional textual information (to more closely link the reference with the text) or removed/replaced a particular citation. These small changes can be seen throughout the manuscript, particularly in the Introduction and Discussion (see marked version of the manuscript).

2.) The figure should be explained in more detail to make it easy to understand.

We have now added a stand-alone paragraph describing Table 1 (see p 9, lns 184-189). We also expanded the table/figure note to be clearer about what the colors corresponded to. Here is that paragraph reproduced:

“Table 1 presents a heat map of the associations between Big Five personality traits and sexual fantasy frequencies (both bivariate and after partialling out the other traits, age, and gender). More intensely green colors correspond to more positive correlations. More intensely red colors correspond to more negative correlations. Varying shades of yellow correspond to values between these intensely positive and negative correlations and are typically smaller and closer to zero. Bivariate and partial correlations greater than |.03| were significant at p = .05.”

3.) The conclusions compiled in paragraphs may need to be improved to explain the results of this study briefly and affirm the existence of novelty in this article.

*** I found the strengths of this article, but there are still weaknesses that need to be fixed.

This article is Acceptable in the PLOS ONE Journal.

We have now more succinctly summarized the results of our study and more squarely focused the contribution of our work. We then make a linkage between this knowledge gained with how it can be leveraged to promote informed conversation in a variety of settings. This paragraph can be found on lns 322-335:

“The present study demonstrated that personality meaningfully relates to patterns of sexual fantasy. At the broad trait level, conscientiousness and agreeableness were associated with lower fantasy frequency, whereas neuroticism predicted greater fantasy engagement. Importantly, examining personality facets revealed more specific psychological correlates of fantasy, with higher depressive tendencies and lower responsibility and respectfulness emerging as the most consistent predictors. This facet-level approach offers a more nuanced understanding of who fantasizes and how. This extends past work that has relied primarily on global traits and provides a preliminary assessment of how sexual fantasies and related to novel and more specific characteristics. Because sexual fantasies are common yet highly variable across individuals, identifying personality correlates may help clinicians and educators support more informed, sex-positive conversations that acknowledge differences in sexual thought and expression. Future work should continue refining these associations and examine whether personality dynamics predict changes in fantasy over time or across relational contexts.”

Thank you again for your positive assessment of our manuscript.

Reviewer #4

1.) The study presents the results of original research. Results reported have not been published elsewhere. Experiments, statistics, and other analyses are performed to a high technical standard and are described in sufficient detail. Conclusions are presented in an appropriate fashion and are supported by the data. The article is presented in an intelligible fashion and is written in standard English. The research meets all applicable standards for the ethics of experimentation and research integrity. The article adheres to appropriate reporting guidelines and community standards for data availability.

Thank you for your thorough review of our manuscript. We appreciate your recognition of how we contextualized the results and of our writing. We also feel that our revised concluding paragraph provides a succinct take-away of our findings and the important implications for having critical policy discussions. This paragraph can be found on lns 322-335:

“The present study demonstrated that personality meaningfully relates to patterns of sexual fantasy. At the broad trait level, conscientiousness and agreeableness were associated with lower fantasy frequency, whereas neuroticism predicted greater fantasy engagement. Importantly, examining personality facets revealed more specific psychological correlates of fantasy, with higher depressive tendencies and lower responsibility and respectfulness emerging as the most consistent predictors. This facet-level approach offers a more nuanced understanding of who fantasizes and how. This extends past work that has relied primarily on global traits and provides a preliminary assessment of how sexual fantasies and related to novel and more specific characteristics. Because sexual fantasies are common yet highly variable across individuals, identifying personality correlates may help clinicians and educators support more informed, sex-positive conversations that acknowledge differences in sexual thought and expression. Future work should continue refining these associations and examine whether personality dynamics predict changes in fantasy over time or across relational contexts.”

Thank you again for your positive assessment of our manuscript.

---

## [Editor Report · Decision Letter 2]

30 Dec 2025

Associations between Big Five Personality Traits, Facets, and Sexual Fantasies

PONE-D-25-12929R2

Dear Dr. Chopik,

We’re pleased to inform you that your manuscript has been judged scientifically suitable for publication and will be formally accepted for publication once it meets all outstanding technical requirements.

Kind regards,

Vilfredo De Pascalis

Academic Editor

PLOS One

Additional Editor Comments (optional):

I see that the authors have addressed the very minor suggested changes; thus, the manuscript can now be accepted for publication.
---

## [Editor Report · Acceptance letter]

PONE-D-25-12929R1

PLOS ONE

Dear Dr. Chopik,

I'm pleased to inform you that your manuscript has been deemed suitable for publication in PLOS ONE. Congratulations! Your manuscript is now being handed over to our production team.

Kind regards,

on behalf of

Professor Vittorio Lenzo

Academic Editor

PLOS ONE